# Kosmos-G: Generating Images in Context with Multimodal Large Language Models

**Xichen Pan**[1,2*]  **Li Dong**[1]  **Shaohan Huang**[1]  **Zhiliang Peng**[1]  **Wenhu Chen**[3]  **Furu Wei**[1]

Microsoft Research[1]   New York University[2]   University of Waterloo[3]

[aka.ms/GeneralAI](aka.ms/GeneralAI)

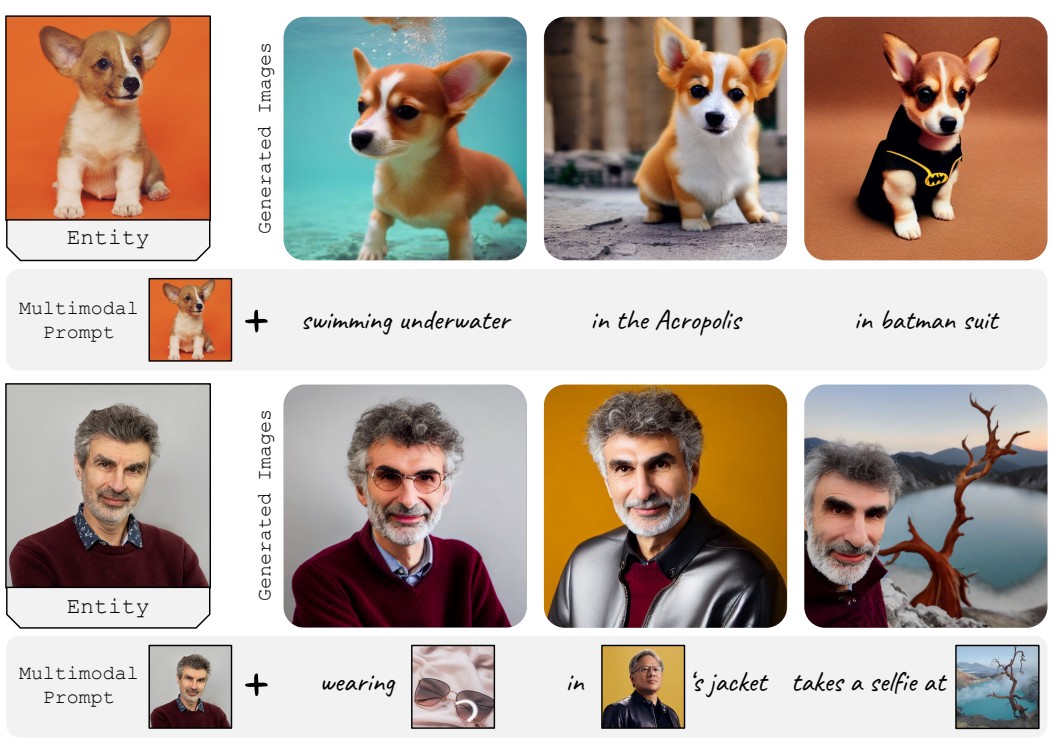

Figure 1: Zero-shot subject-driven generation examples with multimodal prompts. Thanks to the advanced multimodal perception capabilities of MLLM, Kosmos-G can generate high-fidelity subject-driven images by approaching all image inputs as a "foreign language".

## Abstract

Recent advancements in subject-driven image generation have made significant strides. However, current methods still fall short in diverse application scenarios, as they require test-time tuning and cannot accept interleaved multi-image and text input. These limitations keep them far from the ultimate goal of "image as a foreign language in image generation." This paper presents Kosmos-G, a model that leverages the advanced multimodal perception capabilities of Multimodal Large Language Models (MLLMs) to tackle the aforementioned challenge. Our approach aligns the output space of MLLM with CLIP using the textual modality as an anchor and performs compositional instruction tuning on curated data. Kosmos-G demonstrates an impressive capability of zero-shot subject-driven generation with interleaved multi-image and text input. Notably, the score distillation instruction tuning requires no modifications to the image decoder. This allows for a seamless substitution of CLIP and effortless integration with a myriad of U-Net techniques ranging from fine-grained controls to personalized image decoder variants. We posit Kosmos-G as an initial attempt towards the goal of "image as a foreign language in image generation."

---

* Contribution during internship at Microsoft Research.

# 1 INTRODUCTION

In recent studies, advancements in text-to-image (T2I) generation, particularly with diffusion models, have shown remarkable progress in producing highly photorealistic, accurate, and diverse images from textual descriptions. Building on the unprecedented success of producing highly accurate images from text descriptions, numerous studies have delved into more sophisticated subject-driven generation to integrate images into text prompts to generate new customized images.

A group of approaches (Gal et al., 2022; Ruiz et al., 2022; Kumari et al., 2023; Tewel et al., 2023; Avrahami et al., 2023; Hao et al., 2023; Smith et al., 2023) propose to fine-tune the models on each set of reference images, and fail to achieve subject-driven generation through a generalized pre-trained model. Xiao et al. (2023); Wei et al. (2023); Chen et al. (2023; 2022) inject image features into the U-Net of diffusion models. However, such injection methods segregate the guidance for text and images, thereby limiting the effectiveness of joint modeling between the two modalities. Additionally, this approach is challenging to extend to scenarios involving multiple entities. Recent work BLIP-Diffusion (Li et al., 2023a) learns object representations by synthesizing images through the composition of subjects with random backgrounds. This approach effectively endows it with a zero-shot, subject-driven text-to-image generation capability. However, the specific design of its input template and training data restricts its scalability to multiple entities.

In contrast to previous methods that work with original CLIP text encoder Radford et al. (2021), we propose that through leveraging Multimodal Large Language Models (MLLMs) (Alayrac et al., 2022; Hao et al., 2022; Aghajanyan et al., 2022; Huang et al., 2023; Li et al., 2023b), most of the challenges in subject-driven generation may be easily resolved. MLLMs have expanded the perception capabilities of language models to multimodality, enabling them to perceive diverse modalities such as images. The idea of leveraging MLLMs for subject-driven generation presents several advantages: 1) It capitalizes on the inherent vision-language alignment within the MLLM. 2) The MLLM architecture naturally supports interleaved interleaved multi-image and text input. 3) The pre-trained MLLM can effectively model multimodal input in context.

To support zero subject-driven generation with interleaved multi-image and text input, we present KOSMOS-G, which leverages the advanced multimodal perception of MLLM following an "align before instruct" manner. Specifically, we start from the multimodal language modeling stage, leading to the KOSMOS-1 (Huang et al., 2023) MLLM. It envisions language models as a universal task layer, perceiving free-form interleaved vision-language inputs and consolidating various task predictions into textual formats. Given the aligned vision-language representation, we then use the language modality as an anchor and align the output space of the MLLM with the CLIP text encoder. Finally, we perform instruction tuning on the curated data. KOSMOS-G accepts captions as input, where each entity is followed by its segmented image. The model is trained to faithfully reproduce all entities, render the text content, and follow the instructions. In this process, the frozen pre-trained diffusion image decoder serves as a score metric. We distill the learned data distribution to pass the differentiable gradient to the MLLM. This enables KOSMOS-G to harness rich features from the image encoder to generate images faithfully reproducing the contents across various contexts (see Figure 1, we also present examples with more diverse interleaving in Figure 9).

Benefiting from general-purpose pre-training, KOSMOS-G approaches the objective of "image as a foreign language in image generation." This means KOSMOS-G can capture novel concepts from input images and guide personalized creations in a zero-shot setting. Notably, KOSMOS-G also stands as the first model to master zero-shot subject-driven generation with interleaved multi-image and text input. Owing to the score distillation instruction tuning, KOSMOS-G do not need to modify any parameters of the image decoder, i.e., the diffusion U-Net and VAEs. This makes it possible for us to seamlessly substitute CLIP with KOSMOS-G in any image generation system. As a result, a plethora of applications can be unlocked in conjunction with U-Net techniques, ranging from fine-grained controls like ControlNet (Zhang & Agrawala, 2023) to personalized or stylized image decoder variants like amazing community contributed LoRA (Hu et al., 2022) checkpoints.

Overall, we propose KOSMOS-G as an initial attempt towards the objective of "image as a foreign language in image generation." We summarize our main contributions as follows:

1. We propose to leverage the advanced multimodal perception of MLLMs for subject-driven generation with interleaved multi-image and text input.

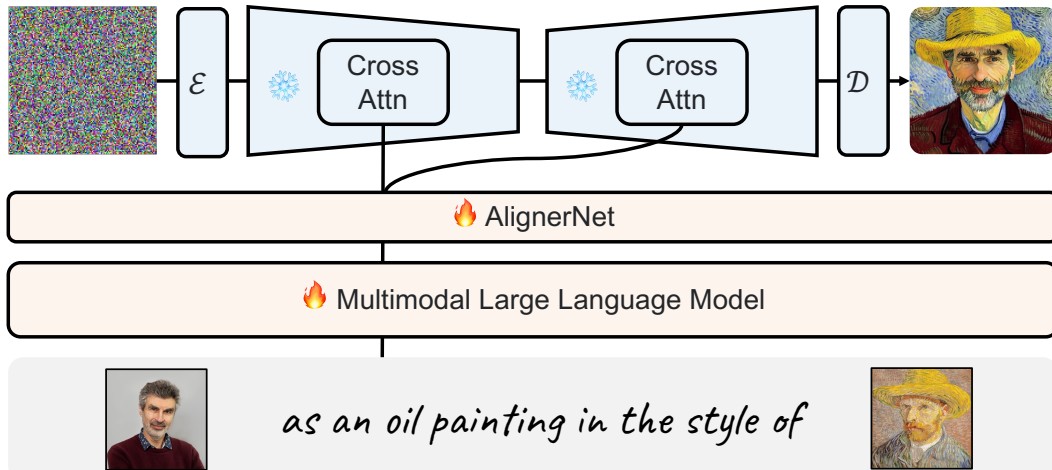

Interleaved Vision-Language Prompt

Figure 2: KOSMOS-G comprises an MLLM for multimodal perception, coupled with an AlignerNet that bridges the MLLM to the diffusion U-Net image decoder. KOSMOS-G can pass the fine concept-level guidance from interleaved input to image decoder, and offer a seamless alternative to CLIP. Orange denotes the trainable modules; Blue denotes the frozen ones.

2. We propose a compositional instruction tuning task, leading to amazing zero-shot multi-entity subject-driven generation capability.

3. Score distillation instruction tuning allows KOSMOS-G to seamlessly interface with a spectrum of U-Net techniques, indicating broad applicability and potential for integration into various frameworks.

## 2 KOSMOS-G: IMAGE AS A FOREIGN LANGUAGE IN IMAGE GENERATION

As shown in Figure 2, KOSMOS-G is a model that can perceive interleaved multi-image and text input and generate subject-driven conditions. Specifically, the backbone of KOSMOS-G MLLM is a Transformer-based causal language model, serving as a general-purpose interface to multimodal input. We train KOSMOS-G following an "align before instruct" manner, the entire training pipeline can be divided into 3 stages:

1. **Multimodal Language Modeling**: We pre-train the MLLM from scratch on multimodal corpora, including monomodal data, cross-modal paired data, and interleaved multimodal data with language modeling loss following KOSMOS-1.

2. **Image Decoder Aligning**: We use the U-Net (Ronneberger et al., 2015) of Stable Diffusion v1.5 (Rombach et al., 2022) as our image decoder. We trained an AlignerNet on only textual data to align the output space of KOSMOS-G to U-Net's input space through CLIP supervision. Here, the language acts as the anchoring modality, ensuring image input is also compatible with the image decoder.

3. **Instruction Tuning**: We further fine-tune KOSMOS-G through a compositional generation task on curated data, with the differentiable gradient passed from the frozen U-Net.

In Stage 1, only the MLLM is trained. In Stage 2, AlignerNet is trained with MLLM frozen. During Stage 3, both AlignerNet and MLLM are jointly trained. The image decoder remains frozen throughout all stages.

### 2.1 MULTIMODAL LANGUAGE MODELING

Following KOSMOS-1, KOSMOS-G perceives general modalities in a unified way. To achieve this, we represent the input format as a single sequence using special tokens. Specifically, we use the

tokens `` and `` to denote start- and end-of-sequence. We also incorporate `<image>` and `</image>` tokens to indicate the start and end of any embedded image representations within the sequence.

Our methodology involves encoding both text tokens and images into vectors, which are then fed into the decoder. For text tokens, we use a lookup table to map them into embeddings. To handle the input images, we employ a vision Transformer (Dosovitskiy et al., 2021) as the embedding module. Furthermore, Resampler (Alayrac et al., 2022) is used as an attentive pooling mechanism to reduce the number of image embeddings. After obtaining the embeddings of an input sequence, we feed them into the Transformer-based decoder. The left-to-right causal decoder processes the sequence in an auto-regressive manner. A softmax classifier on the Transformer is used to assign probabilities to each token in the vocabulary.

KOSMOS-G is first trained using the next-token prediction task. The training objective is to maximize the log-likelihood of tokens in examples. It's important to note that the training loss only takes into account discrete tokens, specifically text tokens. The MLLM component has 24 layers with 2,048 hidden dimensions, 8,192 FFN intermediate size, and 32 attention heads. For faster convergence, the image representation is obtained from a pre-trained CLIP ViT-L/14 model with 1,024 feature dimensions. The images are preprocessed into 224×224 resolution during training. We freeze the parameters of the CLIP model except for the last layer during training. The total number of parameters of the MLLM is about 1.6B.

## 2.2 IMAGE DECODER ALIGNING

After undertaking multimodal language modeling, we have successfully aligned vision and language perception within MLLM. To make KOSMOS-G capable of image generation, we incorporate diffusion models (Sohl-Dickstein et al., 2015) as our image decoder. Specifically, we adopt the widely accepted Stable Diffusion v1.5 (Rombach et al., 2022). It's important to note that we only replace the CLIP text encoder (Radford et al., 2021) with multimodal KOSMOS-G, without making any modifications to the U-Net architecture or weight. This setup allows KOSMOS-G to effectively collaborate with techniques applied to the U-Net, like ControlNet (Zhang & Agrawala, 2023) and various community LoRA (Hu et al., 2022) variants. In this section, we will provide brief preliminaries of latent diffusion models, and then delve into the process of aligning the output space of KOSMOS-G with the image decoder after the aforementioned replacement.

**Preliminaries of Latent Diffusion Models** Diffusion models define a Markov chain of forward diffusion process $q$, adding Gaussian noise samples to the initial real data $\mathbf{z}_0 \sim q(\mathbf{z})$ over $T$ steps. Here, $\mathbf{z}$ denotes latent representations rather than pixel values. The efficient, low-dimensional latent space is approximately perceptually equivalent to high-dimensional RGB space, while the redundant semantically meaningless information present in the pixel domain is eliminated. Perceptual compression models (i.e., VQ-VAE) consisting of $\mathcal{E}$ and $\mathcal{D}$ encode the real data into the latent space and reverse, such that $\mathcal{D}(\mathcal{E}(\mathbf{x})) \approx \mathbf{x}$. Latent diffusion models use latent representations $\mathbf{z} = \mathcal{E}(\mathbf{x})$ instead of working directly with pixel values during the diffusion process. The final output can be decoded back to pixel space via $D(\mathbf{z})$. The separate mild perceptual compression stage only eliminates imperceptible details, leading to competitive generation results with a much lower cost. The forward process $q(\mathbf{z}_t|\mathbf{z}_{t-1})$ at each time step $t$ can be expressed as follows:

$$q(\mathbf{z}_t|\mathbf{z}_{t-1}) = \mathcal{N}(\mathbf{z}_t; \sqrt{1-\beta_t}\mathbf{z}_{t-1}, \beta_t\mathbf{I})$$
$$q(\mathbf{z}_{1:T}|\mathbf{z}_0) = \prod_{t=1}^{T} q(\mathbf{z}_t|\mathbf{z}_{t-1}) \tag{1}$$

in which $\beta_t \in (0, 1)$ denotes the step size. Note $\beta_{t-1} < \beta_t$.

Diffusion models learn a U-Net (Ronneberger et al., 2015) denoted as $\boldsymbol{\epsilon}_\theta$ to reverse the forward diffusion process, constructing desired data samples from the noise. Let $\alpha_t = 1 - \beta_t$ and $\bar{\alpha}_t = \prod_{i=1}^{t} \alpha_i$. We can reparameterize the denoising process $p(\mathbf{z}_{t-1}|\mathbf{z}_t)$ also as a Gaussian distribution. This distribution can be estimated by $\boldsymbol{\epsilon}_\theta$ and takes the following form:

$$p_\theta(\mathbf{z}_{t-1}|\mathbf{z}_t) = \mathcal{N}(\mathbf{z}_{t-1}; \boldsymbol{\mu}_\theta(\mathbf{z}_t, t), \boldsymbol{\Sigma}_\theta(\mathbf{z}_t, t))$$
$$\text{with} \quad \boldsymbol{\mu}_\theta(\mathbf{z}_t, t) = \frac{1}{\sqrt{\alpha_t}}(\mathbf{z}_t - \frac{\beta_t}{\sqrt{1-\bar{\alpha}_t}}\boldsymbol{\epsilon}_\theta(\mathbf{z}_t, t)) \tag{2}$$

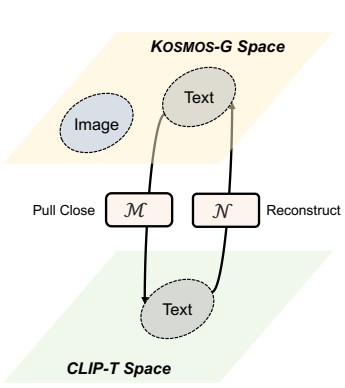

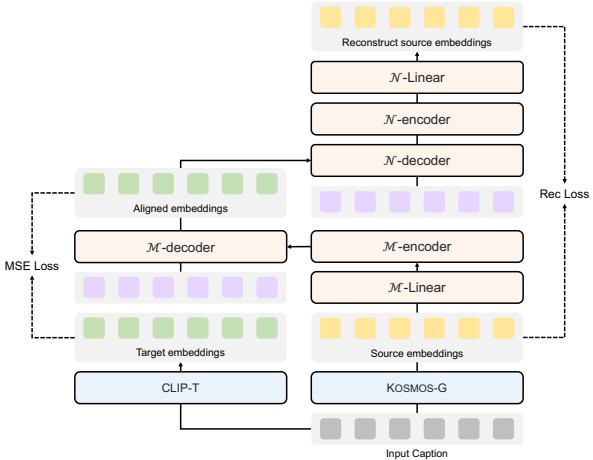

(a) Align process. Text serves as an anchor, image embeddings are naturally aligned throughout the process.

(b) AlignerNet architecture. The Linear layers are used to project the output dimension of MLLM to $d = 768$, the purple elements denote the learned latent queries $\mathbf{Q}_{\mathcal{M}}$ and $\mathbf{Q}_{\mathcal{N}}$.

Figure 3: Overview of alignment.

The learning objective of diffusion models is to approximate the mean $\boldsymbol{\mu}_\theta(\mathbf{z}_t, t)$ in the reverse diffusion process. To achieve this, we can utilize the variational lower bound (ELBO) (Kingma & Welling, 2014) to minimize the negative log-likelihood of $p_\theta(\mathbf{z}_0)$ (Ho et al., 2020). The simplified objective can be expressed as a denoising objective:

$$\mathcal{L}_{diff} = \mathbb{E}_{\mathbf{z}_0, \boldsymbol{\epsilon} \sim \mathcal{N}(0,1), t}\left[\|\boldsymbol{\epsilon} - \boldsymbol{\epsilon}_\theta(\mathbf{z}_t, t)\|^2\right] \tag{3}$$

During inference, (Ho & Salimans, 2022) proposes to use classifier-free guidance to obtain more relevant generation results.

$$\hat{\boldsymbol{\epsilon}} = w \cdot \boldsymbol{\epsilon}_\theta(\mathbf{z}_t, \varphi, t) - (w - 1) \cdot \boldsymbol{\epsilon}_\theta(\mathbf{z}_t, t) \tag{4}$$

where $w$ is guidance scale, $\varphi$ denotes the condition.

**Align Output Space with Diffusion Models** Upon replacing the previous CLIP text encoder with KOSMOS-G, the main focus is to address the misalignment issue between the KOSMOS-G and the image decoder. We discovered that simply fine-tuning KOSMOS-G using the gradient passed from the image decoder results in both trivial alignment and compromised image quality.

Inspired by (Qin et al., 2023), we propose the AlignerNet consisting of an encoder $\mathcal{M}$ and a decoder $\mathcal{N}$ to learn the alignment between the KOSMOS-G source space $\mathbf{S}$ and CLIP text encoder target space $\mathbf{T}$. Given a single text-only caption $\mathbf{C}$, KOSMOS-G source encoder and CLIP text target encoder encode the caption into embeddings denoted as $\mathbf{s} \in \mathbb{R}^{l_s \times d_s}$ and $\mathbf{t} \in \mathbb{R}^{l_t \times d_t}$, respectively. Here, $l$ and $d$ indicate the length of features and embedding dimensions.

As shown in Figure 3a, we employ the encoder $\mathcal{M}$ to minimize the distance between the text source embedding and the target embedding, aiming for a close approximation $\mathcal{M}(\mathbf{s}) \approx \mathbf{t}$ through:

$$\mathcal{L}_{mse} = \mathbb{E}_{\mathbf{s} \sim \mathbf{S}, \mathbf{t} \sim \mathbf{T}}\left[\|\mathbf{t} - \mathcal{M}(\mathbf{s})\|_2^2\right] \tag{5}$$

To mitigate the reduction in feature discrimination, we also employ a decoder $\mathcal{N}$ to reconstruct the source embedding $\mathcal{N}(\mathcal{M}(\mathbf{s})) \approx \mathbf{s}$ through:

$$\mathcal{L}_{rec} = \mathbb{E}_{\mathbf{s} \sim \mathbf{S}}\left[\|\mathbf{t} - \mathcal{N}(\mathcal{M}(\mathbf{s}))\|_2^2\right] \tag{6}$$

Different from (Qin et al., 2023), KOSMOS-G is a vision-language multimodal encoder. The language modality serves as an anchor throughout the process, aligning the entire KOSMOS-G space with the image decoder input space, thus also achieving semantic alignment for the image embeddings.

To efficiently process lengthy sequences consisting of multiple images and minimize memory usage, KOSMOS-G encodes the interleaved vision-language input sequence into variable-length embeddings.

However, the use of variable length embeddings makes the MLP-based GlueNet (Qin et al., 2023) unsuitable for learning alignment. To address this, we employ a Transformer-based architecture in AlignerNet, enabling it to effectively align the source and target spaces with mismatched sequence lengths and embedding dimensions.

As shown in Figure 3b, both $\mathcal{M}$ and $\mathcal{N}$ share a similar architecture design, consisting of a Transformer encoder and a Transformer decoder. The Transformer encoder and decoder in both models comprise 12 layers, with an input dimension $d = 768$ and a hidden dimension of 3072. This configuration results in approximately 225M parameters in total. In the cross attention module of Transformer decoder, we use variable length learned latent queries $\mathbf{Q}_{\mathcal{M}} \in \mathbb{R}^{l_t \times d}$ in $\mathcal{M}$ and $\mathbf{Q}_{\mathcal{N}} \in \mathbb{R}^{l_s \times d}$ in $\mathcal{N}$ to match sequence length. Note that as discussed in Section 4.3, we can still align MLLM with Kosmos-G through directly using diffusion loss in Equation 3 with the help of AlignerNet. While it is more costly and leads to worse performance under the same GPU days.

## 2.3 INSTRUCTION TUNING

After achieving a semantic alignment between KOSMOS-G and the image decoder, our model can successfully generate images following interleaved vision-language guidance. However, the multimodal language modeling and text-only alignment stage only preserve the semantic consistency between the input and output, KOSMOS-G still can not leverage rich features extracted from the image encoder to generate images faithfully reproducing the contents in various contexts.

To pursue our objective of "image as a foreign language in image generation," we curate interleaved vision-language data and use the diffusion loss in Equation 3 to further fine-tune KOSMOS-G. Specifically, we propose a compositional generation task in which we input captions containing entities, with each of them followed by their corresponding images, like "`` *A cat* `<image>` image embedding of the cat `</image>` *and a dog* `<image>` image embedding of the dog `</image>` *sleeping in the garden* `<image>` image embedding of the garden `</image>` ``". Our model is trained to generate images following the input instruction.

To construct the requisite data, we first caption the image, then extract the entities from the caption, and obtain the segmentation results from the image itself. A detailed introduction of the entire pipeline can be found in Section 3.1. Additionally, we leverage the data constructed by (Brooks et al., 2023) for InstructPix2Pix to improve KOSMOS-G's image editing capability. This data is structured as: "`` *caption* `<image>` embedding of the original image `</image>` *edit instruction* ``". We also mix some text-to-image data to preserve the language alignment already achieved.

Our goal is to leverage MLLMs to model image distributions through direct latent space sampling. In this setup, the pre-trained frozen Stable Diffusion U-Net serves as a score metric, distilling the learned data distribution. This strategy is similar to Score Distillation Sampling (SDS) (Poole et al., 2022). From the perspective of score distillation, the KL divergence between KOSMOS-G (denoted as $\phi$, which encodes inputs into condition $\mathcal{C}$) and the score function is equivalently minimized for distilling learned probability density in the image decoder:

$$\min_{\phi} \mathcal{L}_{Diff} = \mathbb{E}_{\mathbf{z}_0, t, \mathcal{C}} \left[ D_{\mathrm{KL}} \big( q(\mathbf{z}_{t-1} | \mathbf{z}_t, \mathbf{z}_0) \, \| \, p_{\theta}(\mathbf{z}_{t-1} | \mathbf{z}_t; \mathcal{C}) \big) \right] \tag{7}$$

This enables KOSMOS-G to leverage rich features from the image encoder to generate an image faithfully reproducing the contents across various contexts. More details about the score distillation instruction tuning can be found in Appendix C.

## 3 MODEL TRAINING

### 3.1 MULTIMODAL TRAINING DATA

The multimodal language modeling stage in Section 2.1 using the same setting of KOSMOS-1 (Huang et al., 2023), where the models are trained on web-scale multimodal corpora, consisted of text corpora, image-caption pairs, and interleaved data of images and texts. For the image decoder aligning stage in Section 2.2, we only use the caption from image-caption pairs. For the instruction tuning stage in Section 2.3, we use constructed data from Open Images V7 dataset (Kuznetsova et al., 2020), the image-caption pairs, as well as the image editing data from InstructPix2Pix (Brooks et al., 2023).

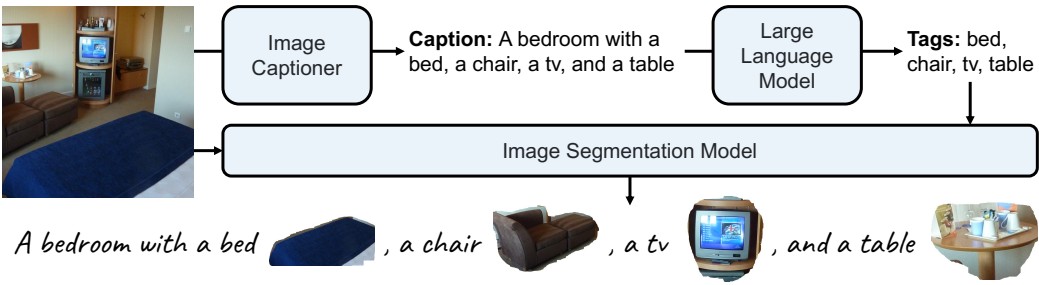

Figure 4: Overview of our data construction pipeline for compositional generation instruction tuning.

**Captions** The image-caption pairs are sourced from multiple datasets, including English LAION-2B (Schuhmann et al., 2022), LAION-400M (Schuhmann et al., 2021), COYO-700M (Byeon et al., 2022), and Conceptual Captions (Sharma et al., 2018; Changpinyo et al., 2021). English LAION-2B, LAION-400M, and COYO-700M are collected from Common Crawl web data by extracting images and the corresponding alt-texts. Conceptual captions are also derived from web pages.

**Constructed Data** We use approximately 9M images from the Open Images V7 dataset (Kuznetsova et al., 2020) to construct our compositional generation instruction tuning data. As illustrated in Figure 4, we begin by generating captions with BLIP-2-OPT-6.7b (Li et al., 2023b). Subsequently, we employ an LLM MPT-7B-Instruct (Team et al., 2023) to extract entities from the captions. The original image, along with the text of each entity, is then input into the text-prompted segmentation model CLIPSeg (Lüddecke & Ecker, 2022) to derive the corresponding image of each entity.

### 3.2 TRAINING SETUP

Our implementation is based on the TorchScale (Ma et al., 2022) library, which is designed for large-scale model training. Following KOSMOS-1 (Huang et al., 2023), we also use MAGNETO (Wang et al., 2022), a Transformer variant, as the backbone architecture of our MLLM and AlignerNet. The whole training process took around four days with 256 NVIDIA V100 GPUs, i.e., one day for image decoder aligning, and three days for instruction tuning. In the instruction tuning stage, we use a blend of constructed data, InstructPix2Pix data, and caption data in a ratio of 2:2:1. For constructed data, to enhance input robustness, we randomly drop the texts of entities with a probability of 0.5 and also maintain the background of the segmented entities with a 0.5 probability. Other training configurations can be found in Appendix A

## 4 EVALUATION

### 4.1 MAIN QUALITATIVE RESULTS

As shown in Figure 5, KOSMOS-G delivers impressive zero-shot generation results across diverse settings, yielding meaningful and coherent outputs even for highly customized subjects. The visual samples showcase generative capabilities in re-contextualization, stylization, modification, and accessory incorporation. Notably, multi-entity subject-driven generation is very challenging even for fine-tuning methods like DreamBooth (Ruiz et al., 2022; Avrahami et al., 2023). While owing from the novel compositional generation instruction tuning, KOSMOS-G is the first model that is capable of achieving this in a zero-shot setting.

### 4.2 QUANTITATIVE RESULTS

We do quantitative evaluations of KOSMOS-G on DreamBench (Ruiz et al., 2022) for single-entity subject-driven generation and MS-COCO (Lin et al., 2014) for text-to-image generation.

The DreamBench dataset contains 30 subjects and features 25 prompt templates, resulting in 750 unique prompts covering skills like re-contextualization, modification, accessorization, etc. We follow prior work to generate 4 images for each prompt to form the 3000 images for a comprehensive evaluation. We follow DreamBooth to adopt DINO, CLIP-I to evaluate the subject fidelity, and CLIP-T to evaluate the text fidelity. We use a classifier-free guidance scale of 7.5 and 100 DPM-Solver (Lu

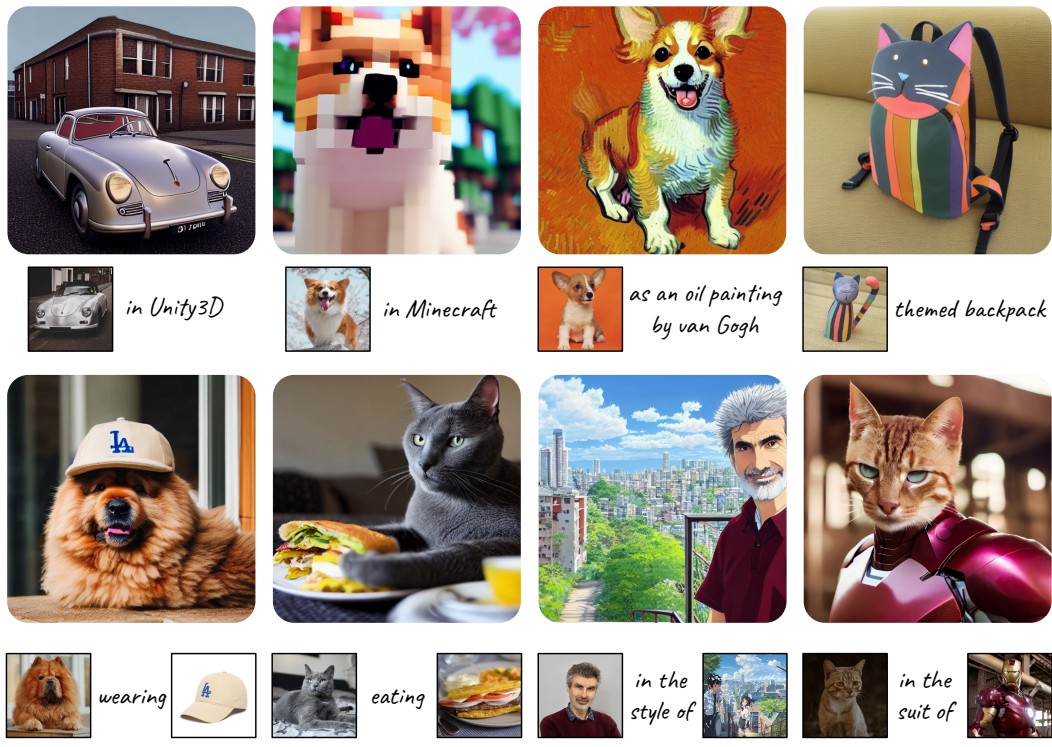

Figure 5: Zero-shot image generation examples with multimodal prompts.

| Methods | DINO↑ | CLIP-I↑ | CLIP-T↑ |
|---|---|---|---|
| Real Images (Oracle) | 0.774 | 0.885 | - |
| *Fine-Tuning* | | | |
| Textual Inversion (Gal et al., 2022) | 0.569 | 0.780 | 0.255 |
| DreamBooth (Ruiz et al., 2022) | 0.668 | 0.803 | 0.305 |
| BLIP-Diffusion (Li et al., 2023a) | 0.670 | 0.805 | 0.302 |
| *Test Time Tuning Free* | | | |
| Re-Imagen* (Chen et al., 2022) | 0.600 | 0.740 | 0.270 |
| SuTI (Chen et al., 2023) | 0.741 | 0.819 | 0.304 |
| BLIP-Diffusion* (Li et al., 2023a) | 0.594 | 0.779 | 0.300 |
| KOSMOS-G* (single image input) | 0.694 | 0.847 | 0.287 |

| Methods | FID↓ |
|---|---|
| *T2I Models* | |
| GLIDE (Nichol et al., 2022) | 12.24 |
| Make-A-Scene (Gafni et al., 2022) | 11.84 |
| DALL-E 2 (Ramesh et al., 2022) | 10.39 |
| SD v1.5$^{\dagger}$ (Rombach et al., 2022) | 9.34 |
| Imagen-3.4B (Saharia et al., 2022) | 7.27 |
| *CLIP-Aligned VL2I Models* | |
| GILL-8B (Koh et al., 2023) | 12.20 |
| Emu-14B (Sun et al., 2023) | 11.66 |
| KOSMOS-G-1.9B | 10.99 |

Table 1: **Left**: Quantitative comparisons on DreamBench. * denotes zero-shot methods. **Right**: Zero-shot FID comparisons on MS-COCO. $^{\dagger}$ indicates results evaluated by us under same settings and seed with KOSMOS-G.

et al., 2022) inference steps for sampling. As shown in Table 1, zero-shot KOSMOS-G outperforms Textual Inversion and Re-Imagen and exhibits marginally better performance than DreamBooth and BLIP-Diffusion with only a single image input. Furthermore, Our results are also comparable with SuTI, without requiring expensive apprenticeship learning supervision. KOSMOS-G accepts only a single image as input, we select a clear image from the 4-7 provided images for each subject to avoid occlusion. We slightly modify the prompt template to ensure better alignment with the instruction tuning data. The images and prompt used can be found in Appendix B.

For the text-to-image generation, We generate images using 30,000 randomly sampled captions from the MS-COCO (2014) validation set. We use a classifier-free guidance scale of 3.0 and 250 DDIM (Song et al., 2021) inference steps for sampling. As shown in Table 1, KOSMOS-G surpasses other CLIP-aligned VL2I models, delivering the optimal alignment results.

## 4.3 ABLATION STUDIES

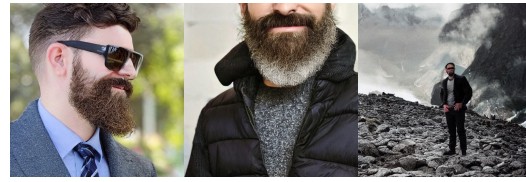 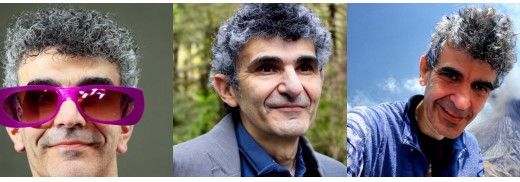

KOSMOS-G before instruction tuning    Stable Diffusion v1.5 Rombach et al. (2022)

Figure 6: Comparisons with cases presented in the second row of Figure 1.

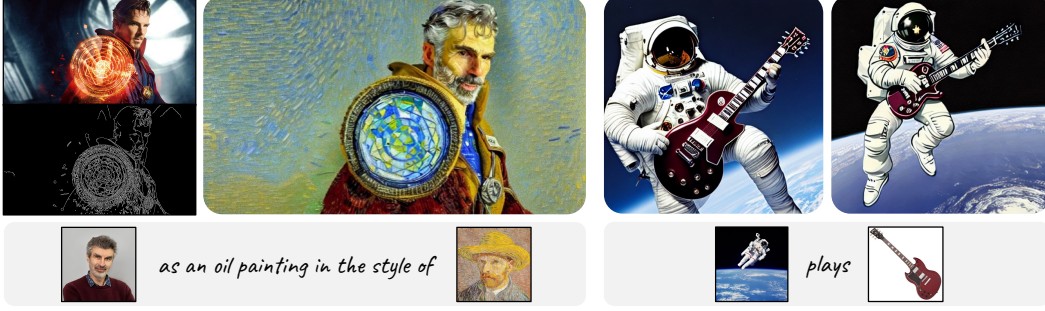

(a) KOSMOS-G with canny control using ControlNet.    (b) KOSMOS-G with LoRA variant.

Figure 7: Various Applications of KOSMOS-G in Conjunction with U-Net Techniques. In Figure 7b, the left image is generated using standard U-Net, the right one is produced with LoRA-tuned U-Net.

We conduct ablation studies to find out the importance of the image decoder aligning and instruction tuning. Table 2 demonstrates that direct end-to-end fine-tuning or using decoder-only architecture fail to generate meaningful images. Incorporating AlignerNet and CLIP supervision, however, results in outcomes close to the original SD v1.5. End-to-end training is also feasible with AlignerNet, but it is more costly due to the additional computational of the U-Net. This leads to worse performance within the same GPU days. We also compared the generation results from KOSMOS-G before instruction tuning and the standard SD v1.5 against our final model. As illustrated in Figure 6, without instruction tuning, KOSMOS-G can only generate contents semantically aligned with the vision-language input. SD baseline also remains at the semantic level and fails to faithfully reproduce the entities in the generated images.

| Methods | FID↓ |
|---|---|
| SD v1.5 | 9.34 |
| E2E w/o AlignerNet | Failed |
| E2E w/ AlignerNet | 11.30 |
| 12-Layers Decoder | Failed |
| 12-Layers AlignerNet | 9.89 |
| 24-Layers AlignerNet | 9.55 |

Table 2: Ablation study results for image decoder aligning on MS-COCO.

### 4.4 APPLICATIONS

As highlighted in Section 2.3, KOSMOS-G can seamlessly replace CLIP in any image generation system. This remarkable property unlocks a myriad of brand-new applications that have never been possible before. We demonstrate its integration with ControlNet (Zhang & Agrawala, 2023) and LoRA variants (Hu et al., 2022) in Figure 7. KOSMOS-G works perfectly with these techniques. Building on the CLIP space, we believe our model will push forward the transition from text-conditioned generation toward vision-language generation, paving the way for numerous novel applications.

## 5 CONCLUSION

We propose KOSMOS-G, a model capable of high-fidelity zero-shot subject-driven generation from interleaved multi-image and text input. Our approach hinges on a unique "align before instruct" pre-training strategy. KOSMOS-G demonstrates competitive single-entity subject-driven image generation and text-to-image capability, it also stands as the first model to extend zero-shot subject-driven image generation to multi-entity scenarios. Furthermore, KOSMOS-G allows seamless replacement of CLIP, unlocking various new applications in conjunction with other U-Net techniques such as ControlNet and LoRA. In general, we present KOSMOS-G as a preliminary effort aimed at achieving the objective of "image as a foreign language in image generation."

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

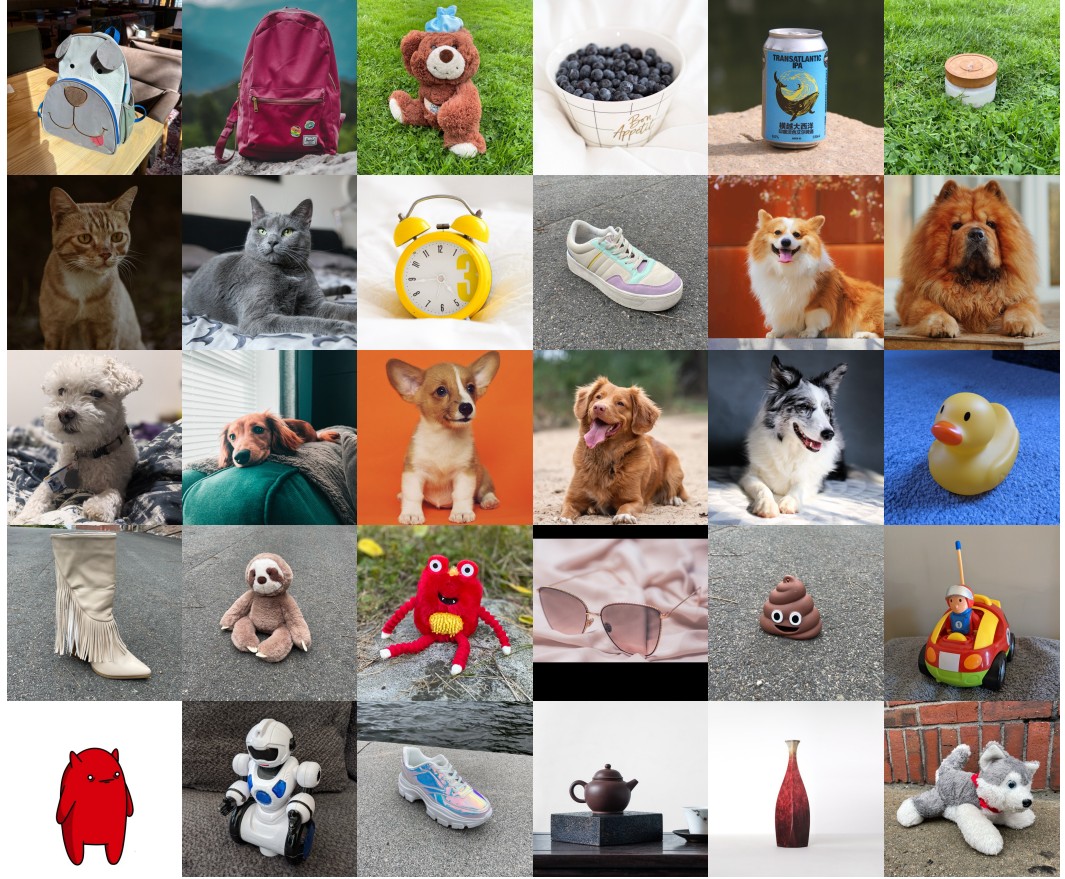

Figure 8: Selected images from DreamBench.

## A    DETAIL TRAINING SETUP

**Multimodal Language Modeling**   We use a batch size of 1.2 million tokens which is broken down as follows: 0.5 million tokens sourced from text corpora, 0.5 million tokens derived from image-caption pairs, and 0.2 million tokens from interleaved data sets. The MLLM is trained for 300,000 steps, corresponding to about 360 billion tokens in total. We adopt the AdamW optimizer with $\beta = (0.9, 0.98)$. Furthermore, we configure the weight decay at 0.01 and the dropout rate at 0.1. The learning rate is set to escalate to 2e-4 during the initial 375 warm-up steps and decay linearly to 0 for the rest of the training steps. For optimization stability, we initiate using Magneto. We use SentencePiece (Kudo & Richardson, 2018) to tokenize the text. We preprocess the data in the "full-sentence" format (Liu et al., 2019), where each input sequence is populated with complete sentences consecutively sampled from one or multiple documents.

**Image Decoder Aligning**   The AlignerNet undergoes training using a batch size of 3,584 sentences for 300,000 steps, with a maximum learning rate of 1e-3. This equates to approximately 1 billion sentences overall. The remaining configurations remain consistent with the previous stage.

**Instruction Tuning**   The MLLM and AlignerNet are jointly trained with a batch size of 1,024 images, totaling approximately 200 million images over 200,000 steps. The learning rate peaks at 1e-3. The rest settings are the same as in the previous stage.

## B    IMAGES AND PROMPTS FOR DREAMBENCH EVALUATION

The input images selected for each entity for DreamBench evaluation are displayed in Figure 8. As depicted in Table 3, We also slightly modified the original prompt to make it align better with the

training data. For the remaining prompt, we simply remove the prefix "a". We observe the prefix will slightly affect the performance of image editing or customized generation. This might be attributed to the high frequency of captions starting with "a photo of" produced by BLIP-2 in our constructed training data. Given that the compositional instruction tuning data does not contain too much editing data when a prompt starts with a prefix like "a", the model often refrains from altering the appearance of the input image. This can be further refined by shifting the data paradigm or by fine-tuning the alignment.

| Original Prompt | Modified Prompt |
|---|---|
| a red {} | {}, red |
| a purple {} | {}, purple |
| a shiny {} | {}, shiny |
| a wet {} | {}, wet |
| a cube shaped {} | {}, cube shaped |

Table 3: Prompt Modification

## C  ADDITIONAL DETAILS ABOUT SCORE DISTILLATION INSTRUCTION TUNING

During the instruction tuning stage, we still utilize the diffusion loss for model training. However, this loss can also be regarded as score distillation, and it is effectively equivalent to optimizing the diffusion loss as the Score Distillation Sampling (SDS) (Poole et al., 2022) loss.

Our objective is to distill the learned score function from the Stable Diffusion U-Net into KOSMOS-G. This process enables KOSMOS-G to encode image features into embeddings that the Stable Diffusion model can understand for subject-driven generation. Essentially, this approach is like pre-training a generalized textual inversion Gal et al. (2022) model, with all conditions learnable by model-seeking. Consider the KOSMOS-G model, denoted as $\phi$, which takes an input $\mathbf{x}$ and produces an output $\mathcal{C} = \phi(\mathbf{x})$. Alongside this, we have the Diffusion U-Net, represented as $\theta$. In our process, we optimize the KOSMOS-G model $\phi$ using the diffusion loss, while keeping the parameters of the Diffusion U-Net $\theta$ frozen. The diffusion loss is expressed as follows:

$$\mathcal{L}_{diff}(\phi) = \mathbb{E}_{\mathbf{z}_0, \boldsymbol{\epsilon} \sim \mathcal{N}(0,1), t}\left[w(t)\|\boldsymbol{\epsilon}_\theta(\mathbf{z}_t; \mathcal{C}, t) - \boldsymbol{\epsilon}\|^2\right] \tag{8}$$

Consider the gradient of $\mathcal{L}_{diff}$:

$$\nabla_\phi \mathcal{L}_{diff}(\phi) = \mathbb{E}_{\mathbf{z}_0, \boldsymbol{\epsilon} \sim \mathcal{N}(0,1), t}\left[w(t) \underbrace{(\boldsymbol{\epsilon}_\theta(\mathbf{z}_t; \mathcal{C}, t) - \boldsymbol{\epsilon})}_{\text{Noise Residual}} \underbrace{\frac{\partial \boldsymbol{\epsilon}_\phi(\mathbf{z}_t; \mathcal{C}, t)}{\mathcal{C}}}_{\text{U-Net Jacobian}} \underbrace{\frac{\partial \mathcal{C}}{\partial \phi}}_{\text{KOSMOS-G Jacobian}}\right] \tag{9}$$

Following the approach in Dreamfusion Poole et al. (2022), we can simplify this equation by omitting certain terms, leading to the SDS loss for KOSMOS-G:

$$\nabla_\phi \mathcal{L}_{SDS}(\phi) = \mathbb{E}_{\mathbf{z}_0, \boldsymbol{\epsilon} \sim \mathcal{N}(0,1), t}\left[w(t) (\boldsymbol{\epsilon}_\theta(\mathbf{z}_t; \mathcal{C}, t) - \boldsymbol{\epsilon}) \frac{\partial \mathcal{C}}{\partial \phi}\right] \tag{10}$$

As established in Dreamfusion Poole et al. (2022), optimizing the SDS loss is effectively equivalent to optimizing the diffusion loss when the U-Net $\theta$ is frozen. From the perspective of score distillation, when using diffusion loss, the KL divergence defined by conditions and the pre-learned score function is equivalently minimized for distilling learned probability density in conditional image synthesis Dong et al. (2023); Luo (2022):

$$\min_\phi \mathcal{L}_{Diff}(\phi) = \mathbb{E}_{\mathbf{z}_0, t, \mathcal{C}}\left[D_{\text{KL}}\big(q(\mathbf{z}_{t-1}|\mathbf{z}_t, \mathbf{z}_0) \,\|\, p_\theta(\mathbf{z}_{t-1}|\mathbf{z}_t; \mathcal{C})\big)\right] \tag{11}$$

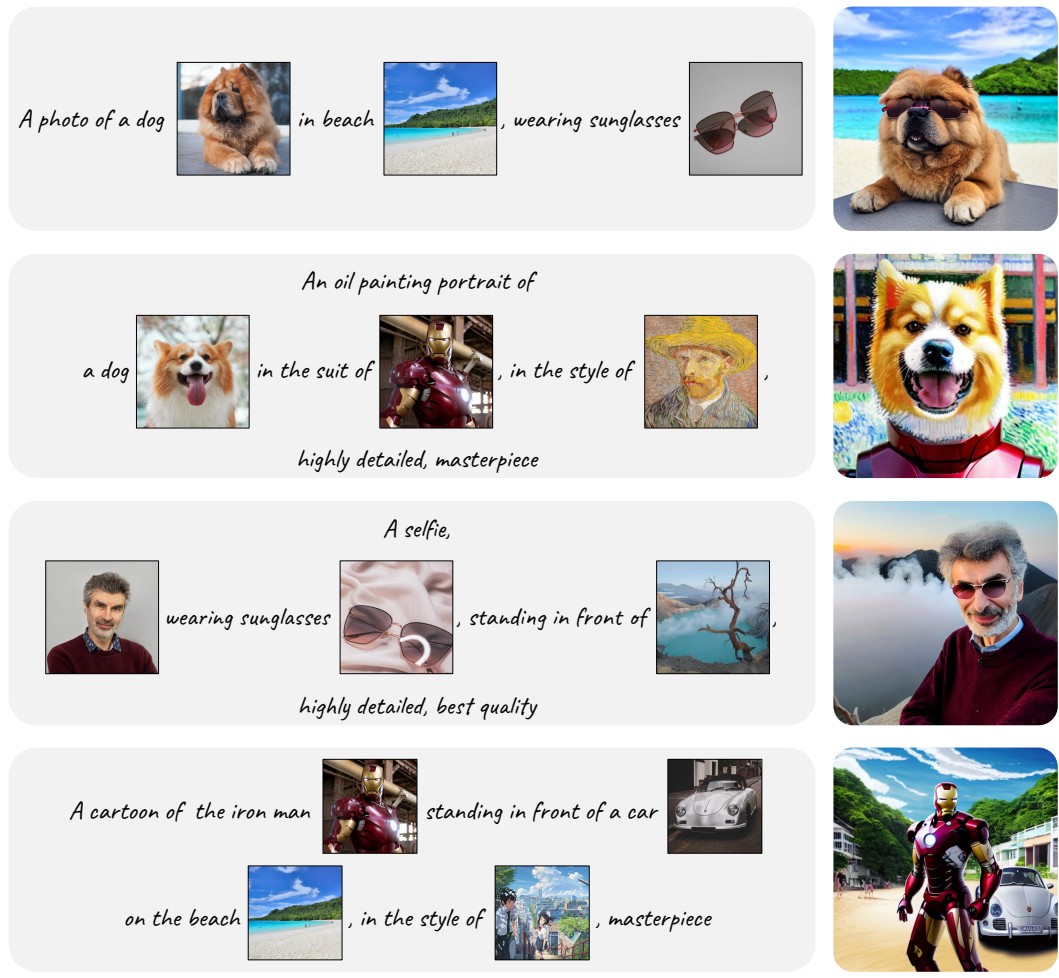

Figure 9: Additional examples under challenging multi-image and text interleaving scenario. These cases show that with KOSMOS-G, users can prompt Stable Diffusion Rombach et al. (2022) by approaching all image inputs as a "foreign language".

## D ADDITIONAL EXAMPLES

In Figure 9, we present cases with more diverse and complex multi-image (3 to 4) and text interleaving scenarios. These examples demonstrate KOSMOS-G's robust performance in handling these challenging cases.

