# OpenReview forum: "Kosmos-G: Generating Images in Context with Multimodal Large Language Models"
_ICLR.cc/2024/Conference — ICLR 2024 poster_

### Official Review · Reviewer_J1rb · 2023-11-01

**Soundness:** 2 fair
**Presentation:** 3 good
**Contribution:** 3 good
**Rating:** 6
**Confidence:** 4

**Summary:**

The paper proposes a framework that combines MLLM and SD to perform image generation/editing with multimodal input. To better bridge the MLLM output space and SD input space, AlignerNet is introduced for feature alignment. Additionally, a large-scale object compositional image generation data is collected and used for training.

**Strengths:**

1. The idea of bridging MLLM and SD for versatile image generation is interesting. MLLM naturally can accept both image and text input, which can provide a more diverse signal to the image generation module and therefore enable new applications.
2. The newly collected compositional image generation dataset should be useful to the community.

**Weaknesses:**

1. I don't see much novelty from AlignerNet.  Compared with GlueNet, AlignerNet merely replaces the MLP with encoder-decoder Transformers but they have the same loss and the same domains (both aligning text embedding). AlignerNet is useful from the experiments, but not novel IMO.
2. Since the training data includes the image editing dataset from InstructPix2Pix. A comparison between previous works on image editing benchmarks should also be conducted. Similarly how is kosmos-g compared with GILL in visual storytelling?
3. In AlignerNet, both MSE and REC losses are used. However, no ablation is done about those two losses.
4. In Tab2, it seems the E2E Fine-tuning fails. However, recent works such as BLIP-Diffusion, EMU, and MGIE can successfully connect MLLM with SD via E2E fine-tuning without any specific alignment. Why Kosmos-G's behavior is different from others and relies on additional alignment?

**Questions:**

1. When constructing the compositional generation dataset, what if multiple objects of the same class exist in the same image? Would the corresponding segmentation mask cover multiple instances in the same mask?

-------------- After rebuttal ----------------
Thank the authors for the last-minute efforts. I have raised the rating to 6. Please stick to this new manuscript and the new title in the camera-ready version if accepted, and further improvement in corresponding writing is also encouraged.

---

> ### Author Response · Authors · 2023-11-19
> **Official Response to Reviewer J1rb (Part 1)**
>
> Dear Reviewer J1rb,
>
> Thanks a lot for your valuable questions. We will address your concerns one by one.
>
> **Q1:** AlignerNet has the same loss and same domains as GlueGen, which is not novel.
>
> **A1:** Thanks for your comments that "The idea of bridging MLLM and SD for versatile image generation is interesting. MLLM naturally can accept both image and text input, which can provide a more diverse signal to the image generation module and therefore enable new applications." We acknowledge that AlignerNet may not be novel, it is adopted from GlueNet [1] for alignment and modified to deal with variable-length outputs. However, we would like to emphasize that for Kosmos-G, the key idea is not about how to align MLLM with SD. As shown in Figure 6, the rough semantic alignment is far from achieving our objective of "Image as a foreign language in image generation". Our main contribution is the idea of leveraging the advanced multimodal perception of MLLM for subject-driven generation. Through instruction tuning the MLLM, we can approach images as a foreign language when prompting Stable Diffusion. We also include additional examples with 3 to 4 image inputs and diverse text interleaving in our revised Appendix D and [here (https://i.ibb.co/BstzDqL/challenge.jpg)](https://i.ibb.co/BstzDqL/challenge.jpg), which shows more about this capability.
>
> **Q2:** Lacks the comparison with previous works on image editing, and comparison with GILL on visual storytelling.
>
> **A2:** We utilize the data from InstructPix2Pix to enable attribute and accessory editing, yet it wouldn't be appropriate to compare with InstructPix2Pix on image editing task. Because we are focusing on subject-driven generation, which does not require preserving the layout of the original input, unlike typical image editing task that requires layout consistency. They are completely two tasks for different purposes. For visual storytelling, we argue that Kosmos-G and GILL [2] are instruction fine-tuned for different purposes. Kosmos-G is fine-tuned on a compositional generation dataset for subject-driven generation, while GILL is fine-tuned on Conceptual Captions (CC3M). Considering the composition nature of our data, our model will not be suitable for visual storytelling, while GILL also struggles with subject-driven generation and composition. However, given the aligned multimodal perception, our method possesses the potential to deal with visual storytelling when trained with interleaved images and captions following GILL.
>
> **Q3:** No ablation is done about MSE and REC loss
>
> **A3:** As shown below or in the revised Appendix C, we present a comprehensive set of ablation studies on the COCO-30k, focusing on the model design and supervision methods used for AlignerNet. We observed that using MSE loss alone leads to slightly worse results, which is consistent with the observation from GlueGen [1].
>
> ### Table 1: FID Score $\downarrow$ on COCO-30k with a guidance scale of 3.0: measuring the fidelity of generated images
>
> | Methods                 | MSE + REC | MSE    | Diff   |
> | ----------------------- | --------- | ------ | ------ |
> | Linear                  | Failed    | Failed | Failed |
> | MLP                     | Failed    | Failed | Failed |
> | Perceiver Resampler [2] | Failed    | Failed | Failed |
> | 12-Layers Decoder       | Failed    | -      | -      |
> | 12-Layers AlignerNet    | 9.89      | 10.23  | 11.30  |
> | 24-Layers AlignerNet    | 9.55      | -      | -      |
>
> ### Table 2: CLIP Score $\uparrow$ on COCO-30k with a guidance scale of 3.0: measuring the alignment between text input and image output
>
> | Methods                 | MSE + REC | MSE    | Diff   |
> | ----------------------- | --------- | ------ | ------ |
> | Linear                  | Failed    | Failed | Failed |
> | MLP                     | Failed    | Failed | Failed |
> | Perceiver Resampler [3] | Failed    | Failed | Failed |
> | 12-Layers Decoder       | Failed    | -      | -      |
> | 12-Layers AlignerNet    | 25.48     | 25.31  | 24.22  |
> | 24-Layers AlignerNet    | 25.57     | -      | -      |
>
> **Q4:** Why E2E Fine-tuning fails.
>
> **A4:** Thanks a lot for this important inquiry, we will clarify it here. E2E Fine-tuning in the paper means we directly connect MLLM with SD and e2e fine-tuning the MLLM. As shown in the above table, direct e2e fine-tuning is feasible with the help of AlignerNet. While it is more costly because we need to pass it through U-Net, leading to a worse performance under the same GPU days. So in practice, we choose to add a specific alignment to accelerate training.

---

> ### Author Response · Authors · 2023-11-19
> **Official Response to Reviewer J1rb (Part 2)**
>
> **Q5:** When constructing the compositional generation dataset, what if multiple objects of the same class exist in the same image? Would the corresponding segmentation mask cover multiple instances in the same mask?
>
> **A5:** Thank you for this insightful question. There do exist cases that multiple instances of the same object class appear in a single image. We simply choose the largest connected sub-region. This approach is intuitive as it picks the clearest, unobstructed object for training. It is also consistent with the inference usage, as we represent each object in the prompt using a single image.
>
> ## Reference
>
> [1] Qin, C., Yu, N., Xing, C., Zhang, S., Chen, Z., Ermon, S., ... & Xu, R. (2023). GlueGen: Plug and Play Multi-modal Encoders for X-to-image Generation. *arXiv preprint arXiv:2303.10056*.
>
> [2] Koh, J. Y., Fried, D., & Salakhutdinov, R. (2023). Generating images with multimodal language models. *arXiv preprint arXiv:2305.17216*.
>
> [3] Alayrac, J. B., Donahue, J., Luc, P., Miech, A., Barr, I., Hasson, Y., ... & Simonyan, K. (2022). Flamingo: a visual language model for few-shot learning. *Advances in Neural Information Processing Systems*, *35*, 23716-23736.

---

> > ### Comment · Reviewer_J1rb · 2023-11-23
> > **Re: Official Response**
> >
> > Thank the authors for the detailed response. I am satisfied with the clarification about E2E fine-tuning as well as the main focus of this paper being positioned as `subject-driven generation`.
> >
> > However, the title `GENERATING IMAGES IN CONTEXT WITH MULTIMODAL LARGE LANGUAGE MODELS` is a bit too broad, more than just subject-driven generation. GILL and many other works can also be umbrellaed under this title. Also, many related descriptions in the introduction/abstract need to be revised to more faithfully reflect the uniqueness and main functionality of your work, eg, the caption in Fig1: `It can perceive generalized vision-language inputs that span multiple images and faithfully generate images.` might not be accurate since it cannot conduct faithful image editing and visual story-telling with multiple-image input.
> >
> > Overall, I appreciate the author(s)' efforts and insights in this work. **I would raise my score if the author(s) can revise their manuscript correspondingly.**

---

> > > ### Author Response · Authors · 2023-11-23
> > > **Thanks for Your Response**
> > >
> > > Thanks a lot for your insightful response. We totally agree with the point that we need to make our paper's focus clear (subject-driven generation). We will try to improve our manuscript in last minute. Though it may not be perfect before the deadline, we commit to revising it to incorporate your suggestions. We will ping you after finishing revision.

---

> > > ### Author Response · Authors · 2023-11-23
> > > **Manuscript Updated**
> > >
> > > Dear reviewer J1rb:
> > >
> > > Thanks again for your insightful feedback, we have changed our title to "MULTIMODAL LARGE LANGUAGE MODELS FOR SUBJECT-DRIVEN IMAGE GENERATION" to highlight the main focus of this paper as subject-driven generation. We have also updated our manuscript to reflect your feedback with all changes highlighted using blue color. You can find the changelog in the main thread. As you know the deadlien is really soon, though the manuscript may not be perfect before the deadline, we promise we will further improve it to reflect your feedback. We are still eager to engage in further discussion and address any additional concerns you may have. We would be very grateful if you could raise your rating accordingly.

---

> ### Author Response · Authors · 2023-11-21
> **Reviewer-Author Discussion Period Ends in TWO Days**
>
> Thanks again for reviewing our paper. We hope that we were able to address your concerns in our response. As the deadline is approaching, please let us know if you have any further questions before the reviewer-author discussion period ends. We are glad to address your further concerns.

---

> ### Author Response · Authors · 2023-11-22
> **Final Day for Discussion: We Hope to Have You in Our Discussion**
>
> Thank you again for dedicating your time to reviewing our paper. We understand the discussion time is limited, yet we do hope to have your participation in our Reviewer-Author discussion. It means a lot to us to fully address your concern and improve our paper during the discussion period. Even within the remaining one-day discussion period, we are still happy to know your further concerns and will do our utmost to resolve them.

---

> ### Author Response · Authors · 2023-11-23
> **Final Reminder for Discussion: 7 Hours Left until the Deadline**
>
> Dear reviewer: As the deadline draws near, we kindly request your feedback on our rebuttal. We are eager to engage in further discussion and address any additional concerns you may have.

---

### Official Review · Reviewer_VSX4 · 2023-11-02

**Soundness:** 2 fair
**Presentation:** 3 good
**Contribution:** 3 good
**Rating:** 6
**Confidence:** 4

**Summary:**

This paper explores image generation from generalized vision-language inputs, especially involving multiple images. Named KOSMOS-G, a model that leverages the advanced perception capabilities of MLLMs, and aligns the output space of MLLM with CLIP using the textual modality as an anchor and performs compositional instruction tuning on curated data.

**Strengths:**

The problem of image generation conditioning on generalized vision-language inputs is an interesting problem, and the proposed approach seems to show some promising results.
The idea of aligning KOSMOS-G Space with the CLIP-T Space and then directly leveraging the stable diffusion models seems a valid approach.
The qualitative results show some good capabilities of the proposed method.

**Weaknesses:**

The ablation study seems not very comprehensive, for example, if the goal is to align the two representation spaces, there should be other options to achieve the alignment design, so why is the current AlignerNet design the best, maybe more justification and ablation study are needed here.

**Questions:**

see the weakness part.

---

> ### Author Response · Authors · 2023-11-19
> **Official Response to Reviewer VSX4**
>
> Dear Reviewer VSX4,
>
> Thanks a lot for your valuable feedback and advice.
>
> **Q:** More justification and ablation study for AlignerNet
>
> **A:** In the revised Appendix C, we have expanded the ablation study for AlignerNet on the COCO-30k dataset, focusing on its model design and supervision. We investigated various configurations, including the use of MSE loss, reconstruction loss, and diffusion loss in supervising the aligned network, while keeping other modules such as the MLLM frozen.
>
> The results demonstrate that simple networks such as Linear, MLP, and Resampler fail to achieve satisfactory alignments. Moreover, due to MLLM's variable-length output, MLP-based GlueNet [1] and encoder-only Transformer models are unsuitable for learning alignment. Decoder-only architecture also fails due to its inefficiency in modeling dimension transformations.
>
> For loss functions, using only MSE loss results in worse results, which is consistent with findings from GlueGen [1]. Direct end-to-end training is more costly because we need to pass it through U-Net, leading to worse performance under the same GPU days.
>
> Notably, AlignerNet surpasses the performance of GlueNet+T5. For instance, GlueNet + T5 attained CLIP scores of 20.67, 23.24, and 23.74 on the COCO-5k dataset at guidance scales of 1.5, 5, and 7.5 respectively, and a FID score of 14.32 at a guidance scale of 3.0 on COCO-30k. These results highlight the effectiveness of AlignerNet in achieving superior alignment, validating the soundness of its design and supervision.
>
> ### Table 1: FID Score $\downarrow$ on COCO-30k with a guidance scale of 3.0: measuring the fidelity of generated images
>
> | Methods               | MSE + REC | MSE   | Diff   |
> | --------------------- | --------- | ------ | ------ |
> | Linear                | Failed    | Failed | Failed |
> | MLP                   | Failed    | Failed | Failed |
> | Perceiver Resampler [2] | Failed    | Failed | Failed |
> | 12-Layers Decoder     | Failed | -      | -      |
> | 12-Layers AlignerNet  | 9.89     | 10.23 | 11.30 |
> | 24-Layers AlignerNet  | 9.55     | -      | -      |
>
> ### Table 2: CLIP Score $\uparrow$ on COCO-30k with a guidance scale of 3.0: measuring the alignment between text input and image output
>
> | Methods               | MSE + REC | MSE   | Diff |
> | --------------------- | --------- | ------ | ------ |
> | Linear                | Failed    | Failed | Failed |
> | MLP                   | Failed    | Failed | Failed |
> | Perceiver Resampler [2] | Failed    | Failed | Failed |
> | 12-Layers Decoder     | Failed | -      | -     |
> | 12-Layers AlignerNet  | 25.48   | 25.31 | 24.22 |
> | 24-Layers AlignerNet  | 25.57   | -      | -     |
>
> ## Reference
>
> [1] Qin, C., Yu, N., Xing, C., Zhang, S., Chen, Z., Ermon, S., ... & Xu, R. (2023). GlueGen: Plug and Play Multi-modal Encoders for X-to-image Generation. *arXiv preprint arXiv:2303.10056*.
>
> [2] Alayrac, J. B., Donahue, J., Luc, P., Miech, A., Barr, I., Hasson, Y., ... & Simonyan, K. (2022). Flamingo: a visual language model for few-shot learning. *Advances in Neural Information Processing Systems*, *35*, 23716-23736.

---

> ### Author Response · Authors · 2023-11-21
> **Reviewer-Author Discussion Period Ends in TWO Days**
>
> Thanks again for reviewing our paper. We hope that we were able to address your concerns in our response. As the deadline is approaching, please let us know if you have any further questions before the reviewer-author discussion period ends. We are glad to address your further concerns.

---

> ### Author Response · Authors · 2023-11-22
> **Final Day for Discussion: We Hope to Have You in Our Discussion**
>
> Thank you again for dedicating your time to reviewing our paper. We understand the discussion time is limited, yet we do hope to have your participation in our Reviewer-Author discussion. It means a lot to us to fully address your concern and improve our paper during the discussion period. Even within the remaining one-day discussion period, we are still happy to know your further concerns and will do our best to resolve them.

---

> > ### Comment · Reviewer_VSX4 · 2023-11-22
> >
> > Thank you for the clarifications. I've read other reviews and rebuttals.

---

> > > ### Author Response · Authors · 2023-11-23
> > > **Thanks for Your Response**
> > >
> > > Thank you very much for your reply! We truly value your involvement in this discussion. We are happy to know that our rebuttal has clarified your concern. We would be very grateful if you could raise your rating accordingly.

---

> ### Author Response · Authors · 2023-11-23
> **Manuscript Updated**
>
> Dear reviewer VSX4:
>
> Thanks again for your valuable feedback, we have updated our manuscript to reflect your feedback with all changes highlighted using blue color. You can find the changelog in the main thread. We are still eager to engage in further discussion and address any additional concerns you may have. We would be very grateful if you could raise your rating accordingly.

---

### Official Review · Reviewer_hrQd · 2023-11-09

**Soundness:** 3 good
**Presentation:** 3 good
**Contribution:** 3 good
**Rating:** 6
**Confidence:** 4

**Summary:**

Kosmos-G aligns the outputs of MLLM to the embedding space of CLIP text encoder, which can be fed into Stable Diffusion model for image generation with context of any form.

**Strengths:**

1. The KOSMOS-G's ability to achieve zero-shot multi-entity subject-driven generation is notable. The method addresses an underexplored area in image generation by focusing on generalized vision-language inputs and multiple images, the method leverage existing advancements in both multimodal language models and image generation.

2. By the alignment of the output space of MLLMs with CLIP and Score distillation instruction tuning, KOSMOS-G can achieve subject-driven generation and image editing without any training on diffusion models, highlighting its potential for integration into different frameworks.

**Weaknesses:**

1. The paper repeatedly mentions KOSMOS-G's ability to master zero-shot multi-entity generation and handle interleaved image-text input. However, the practical cases presented in the paper seem to focus primarily on image editing capabilities. Look forward to showing more cases with complex and rich scenarios to further illustrate the capabilities of the model. : 1）the paper only demonstrates cases with a maximum of two images, failing to showcases with more than two images as inputs.  2) the paper predominantly showcases and evaluates image-text-image input scenarios, leaving more diverse multi-image and text interleaving cases unexplored.

2. Section 2.3 discusses the "Score distillation instruction tuning" technique, but the description lacks clarity. The paper should provide a more precise definition of the entities involved in calculating the KL divergence, along with any specific mathematical formulas or equations for better understanding.  In addition, is KL divergence loss necessary? Is it feasible to directly apply diffusion model's loss for training?

3. The paper highlights the exceptional subject-driven generation capabilities of KOSMOS-G, particularly when not training the diffusion model.   I would like to ask if the authors have explored the possibility of further enhancing subject-driven generation, like training the diffusion model.

**Questions:**

See above.

---

> ### Author Response · Authors · 2023-11-19
> **Official Response to Reviewer hrQd (Part 1)**
>
> Dear Reviewer hrQd,
>
> Thanks a lot for your insightful feedback and kind advice. We would like to address your concerns one by one.
>
> **Q1:** The paper does not demonstrate scenarios involving more than two images and lacks exploration of diverse multi-image and text interleaving cases.
>
> **A1:** Thank you for your valuable suggestion. We acknowledge the importance of showcasing more complex scenarios in demonstrating the capabilities of Kosmos-G. It's worth noting that subject-driven generation across multiple images, especially more than two, is challenging. This is true even for state-of-the-art fine-tuning methods like Break-a-Scene [1], which supports up to three subjects. Zero-shot methods like FastComposer [2] and Subject-Diffusion [3] can handle up to two images but struggle with attribute and accessory editing and also require attention map manipulation. Owning to our collected training data, it is possible to treat the image input as a text token and prompt Kosmos-G like Stable Diffusion. This unique feature allows for more diverse and complex multi-image and text interleaving scenarios. We have included additional examples with 3 to 4 image inputs and diverse text interleaving in our revised Appendix D and [here (https://i.ibb.co/BstzDqL/challenge.jpg)](https://i.ibb.co/BstzDqL/challenge.jpg). These examples demonstrate Kosmos-G's robust performance in handling these challenging cases.
>
> **Q2:** "Score distillation instruction tuning" lacks clarity. Is it feasible to directly apply diffusion model's loss for training?
>
> **A2:** Thank you for your question regarding the clarity of "Score Distillation Instruction Tuning". We would like to emphasize that during the instruction tuning stage, we still utilize the diffusion loss for model training. However, this loss can also be regarded as score distillation, and it is effectively equivalent to optimizing the diffusion loss as the SDS loss. We will elaborate on these in the following text.
>
> Our objective is to distill the learned score function from the Stable Diffusion U-Net into Kosmos-G. This process enables Kosmos-G to encode image features into embeddings that the Stable Diffusion model can understand for subject-driven generation. Essentially, this approach is like pre-training a generalized textual inversion [4] model, with all conditions learnable by model-seeking.
>
> Consider the Kosmos-G model, denoted as $\phi$, which takes an input $\mathbf{x}$ and produces an output $\mathcal{C}=\phi(\mathbf{x})$. Alongside this, we have the Diffusion U-Net, represented as $\theta$. In our process, we optimize the Kosmos-G model $\phi$ using the diffusion loss, while keeping the parameters of the Diffusion U-Net $\theta$ frozen. The diffusion loss is expressed as follows:
>
> $$
> \mathcal{L} _{diff}(\phi) = \mathbb{E} _{\mathbf{z} _0, \boldsymbol{\epsilon} \sim \mathcal{N}(0, 1), t} \Big[w(t)\|\boldsymbol{\epsilon} _\theta(\mathbf{z} _t; \mathcal{C}, t) - \boldsymbol{\epsilon}\|^2 \Big]
> $$
>
> consider the gradient of $\mathcal{L} _{diff}$:
> $$
> \nabla _\phi \mathcal{L} _{diff}(\phi) = \mathbb{E} _{\mathbf{z} _0, \boldsymbol{\epsilon} \sim \mathcal{N}(0, 1), t} \Bigg[w(t)
> \underbrace{\left(\boldsymbol{\epsilon} _\theta(\mathbf{z} _t; \mathcal{C}, t)  - \boldsymbol{\epsilon} \right) \vphantom{\partial \mathcal{C} \over \partial \phi}} _{\text{Noise Residual}}
> \underbrace{{\partial \boldsymbol{\epsilon} _\phi(\mathbf{z} _t; \mathcal{C}, t) \over \mathcal{C}} \vphantom{\partial \mathcal{C} \over \partial \phi}} _{\text{U-Net Jacobian}}
> \underbrace{{\partial \mathcal{C} \over \partial \phi}} _{\text{Kosmos-G Jacobian}}\Bigg]
> $$
>
> Following the approach in Dreamfusion [6], we can simplify this equation by omitting certain terms, leading to the SDS loss for Kosmos-G:
> $$
> \nabla _\phi \mathcal{L} _{SDS}(\phi) \triangleq \mathbb{E} _{\mathbf{z} _0, \boldsymbol{\epsilon} \sim \mathcal{N}(0, 1), t} \Bigg[w(t)\left(\boldsymbol{\epsilon} _\theta(\mathbf{z} _t; \mathcal{C}, t) - \boldsymbol{\epsilon} \right) {\partial \mathcal{C} \over \partial \phi}\Bigg]
> $$
>
> As established in Dreamfusion [6], optimizing the SDS loss is effectively equivalent to optimizing the diffusion loss when the U-Net $\theta$ is frozen. From the perspective of score distillation, when using diffusion loss, the KL divergence defined by conditions and the pre-learned score function is equivalently minimized for distilling learned probability density in conditional image synthesis [5, 7]:
> $$
> \mathop{\min}\limits _{\phi} \mathcal{L} _{Diff}(\phi) = \mathbb{E} _{\mathbf{z} _0, t, \mathcal{C}} \Big[D _{\rm{KL}}\big(q(\mathbf{z} _{t-1}|\mathbf{z} _t, \mathbf{z} _{0})~\| ~p _\theta(\mathbf{z} _{t-1}|\mathbf{z} _t; \mathcal{C})\big)\Big]
> $$

---

> ### Author Response · Authors · 2023-11-19
> **Official Response to Reviewer hrQd (Part 2)**
>
> **Q3:** The possibility of further enhancing subject-driven generation, like training the diffusion model.
>
> **A3:** Thank you for your valuable suggestion regarding the enhancement of subject-driven generation through training the diffusion model. We designed Kosmos-G to function as a minimal, plug-and-play module within Stable Diffusion, offering a more community-friendly approach compared to a fully modularized system. This design choice allows Kosmos-G to seamlessly replace CLIP, facilitating easier integration with other U-Net techniques, such as ControlNet and LoRA, as demonstrated in our paper. While we acknowledge that some research, like BLIP-Diffusion [8], involves training the diffusion model, our approach as outlined in Table 1 achieves superior performance without the need to train the U-Net. We agree the subject-driven generation capability can be improved by tuning more parameters. However, our current resources, specifically V100 GPUs with 32GB VRAM, constrain our ability to fully explore the outcomes of training Kosmos-G with an unlocked U-Net.
>
> ## Reference:
>
> [1] Avrahami, O., Aberman, K., Fried, O., Cohen-Or, D., & Lischinski, D. (2023). Break-A-Scene: Extracting Multiple Concepts from a Single Image. *arXiv preprint arXiv:2305.16311*.
>
> [2] Xiao, G., Yin, T., Freeman, W. T., Durand, F., & Han, S. (2023). FastComposer: Tuning-Free Multi-Subject Image Generation with Localized Attention. *arXiv preprint arXiv:2305.10431*.
>
> [3] Ma, J., Liang, J., Chen, C., & Lu, H. (2023). Subject-diffusion: Open domain personalized text-to-image generation without test-time fine-tuning. *arXiv preprint arXiv:2307.11410*.
>
> [4] Gal, R., Alaluf, Y., Atzmon, Y., Patashnik, O., Bermano, A. H., Chechik, G., & Cohen-Or, D. (2022). An image is worth one word: Personalizing text-to-image generation using textual inversion. *arXiv preprint arXiv:2208.01618*.
>
> [5] Dong, R., Han, C., Peng, Y., Qi, Z., Ge, Z., Yang, J., ... & Yi, L. (2023). Dreamllm: Synergistic multimodal comprehension and creation. *arXiv preprint arXiv:2309.11499*.
>
> [6] Poole, B., Jain, A., Barron, J. T., & Mildenhall, B. (2022). Dreamfusion: Text-to-3d using 2d diffusion. *arXiv preprint arXiv:2209.14988*.
>
> [7] Luo, C. (2022). Understanding diffusion models: A unified perspective. *arXiv preprint arXiv:2208.11970*.
>
> [8] Li, D., Li, J., & Hoi, S. C. (2023). Blip-diffusion: Pre-trained subject representation for controllable text-to-image generation and editing. *arXiv preprint arXiv:2305.14720*.

---

> ### Author Response · Authors · 2023-11-21
> **Reviewer-Author Discussion Period Ends in TWO Days**
>
> Thanks again for reviewing our paper. We hope that we were able to address your concerns in our response. As the deadline is approaching, please let us know if you have any further questions before the reviewer-author discussion period ends. We are glad to address your further concerns.

---

> ### Author Response · Authors · 2023-11-22
> **Final Day for Discussion: We Hope to Have You in Our Discussion**
>
> Thank you again for dedicating your time to reviewing our paper. We understand the discussion time is limited, yet we do hope to have your participation in our Reviewer-Author discussion. It means a lot to us to fully address your concern and improve our paper during the discussion period. Even within the remaining one-day discussion period, we are still happy to know your further concerns and will do our utmost to resolve them.

---

> ### Author Response · Authors · 2023-11-23
> **Final Reminder for Discussion: 7 Hours Left until the Deadline**
>
> Dear reviewer: As the deadline draws near, we kindly request your feedback on our rebuttal. We are eager to engage in further discussion and address any additional concerns you may have.

---

> ### Author Response · Authors · 2023-11-23
> **Manuscript Updated**
>
> Dear reviewer hrQd:
>
> Thank you once again for your valuable feedback, we have updated our manuscript to reflect your feedback with all changes highlighted using blue color. You can find the changelog in the main thread. We are still eager to engage in further discussion and address any additional concerns you may have. If you find our response addressed your concern, we would deeply appreciate it if you could consider raising our rating.

---

### Author Response · Authors · 2023-11-19
**General Response to All Reviewers**

Thank you to all reviewers for dedicating your time to review our paper, and for providing valuable and insightful feedback. We are thrilled that the reviewers acknowledge our noteworthy generation results and the novel approach of leveraging MLLM's advanced multimodal perception for image generation.

We've updated our paper to include additional ablation studies and image generation results, which you can find in Appendices C and D, respectively. These updates are intended to address your concerns regarding our ablation studies on AlignerNet and the capabilities of Kosmos-G in more advanced scenarios.

We look forward to hearing from you. If you have any further questions or concerns, please do not hesitate to let us know. We are eager to address them promptly before the discussion deadline.

---

### Author Response · Authors · 2023-11-23
**Updates of Our Manuscript**

Thanks to the feedback of reviewers, we have improved our manuscript a lot (highlighted using blue color in PDF). Here is the change log:
1. As suggested by reviewer J1rb, we have changed our title to "MULTIMODAL LARGE LANGUAGE MODELS FOR SUBJECT-DRIVEN IMAGE GENERATION" to highlight the main focus of this paper being positioned as subject-driven generation.
2. As suggested by reviewer J1rb, we have changed our abstract, introduction, captions, conclusion to more faithfully reflect the uniqueness and main functionality of our work
3. As suggested by reviewer VSX4 and J1rb, we have added additional ablation studied in Appendix C to provide detailed ablation about methods and supervision, we also show that AlignerNet can work with E2E fine-tuning.
4. As suggested by reviewer hrQd, we have included additional examples with 3 to 4 image inputs and diverse text interleaving in our revised Appendix D.
5. As suggested by reviewer hrQd, we have provided a detailed discussion about score distillation instruction fine-tuning in Appendix E.
6. We have also updated several sections to talk about the abovementioned contents.

---

### Meta-Review · Area_Chair_qjyr · 2023-12-06

**Metareview:**

The AC recommends the acceptance of the paper due to its novel approach to image generation, leveraging the potential of multimodal large language models (MLLMs) and the Stable Diffusion model. Reviewers appreciated the novelty of KOSMOS-G in achieving zero-shot multi-entity subject-driven generation, addressing an underexplored area in image generation that focuses on generalized vision-language inputs and handling multiple images. This approach stands out for its capability to align the output space of MLLMs with CLIP and perform compositional instruction tuning, enabling subject-driven generation and image editing without training on diffusion models.

However, reviewers noted some areas for improvement. They pointed out that the paper predominantly showcases image editing capabilities and lacks a demonstration of more complex scenarios with multiple images. Additionally, there were calls for more clarity in explaining the "Score distillation instruction tuning" technique. Despite these areas of concern, the authors responded comprehensively in their rebuttal, addressing each point with additional information and clarifications. They also changed the title of the paper to address this concern.

The reviewers' confidence in the soundness, presentation, and contribution of the paper, combined with the authors' efforts to address feedback, support the decision to accept this submission. The research provides a meaningful advancement in the field of image generation, particularly in leveraging advanced multimodal language models for complex, subject-driven tasks.

**Justification For Why Not Higher Score:**

Each reviewer rated the submission as "6: marginally above the acceptance threshold," indicating that while the paper has merits warranting acceptance, it still has limitation as described in the detailed review and the summary above.

**Justification For Why Not Lower Score:**

All the reviewers unanimously accepted the paper, especially after the rebuttal. No basis to overturn the reviews.

---

### Decision · Program_Chairs · 2024-01-16

Accept (poster)